# Waist Circumference and Healthy Lifestyle Preferences/Knowledge Monitoring in a Preschool Obesity Prevention Program

**DOI:** 10.3390/nu11092139

**Published:** 2019-09-07

**Authors:** Marco Poeta, Rossella Lamberti, Dario Di Salvio, Grazia Massa, Nives Torsiello, Luca Pierri, Anna Pia Delli Bovi, Laura Di Michele, Salvatore Guercio Nuzio, Pietro Vajro

**Affiliations:** Pediatrics, Department of Medicine, Surgery and Dentistry “Scuola Medica Salernitana”, University of Salerno, 84081 Baronissi, Salerno, Italy (M.P.) (R.L.) (D.D.S.) (G.M.) (N.T.) (L.P.) (A.P.D.B.) (L.D.M.) (S.G.N.)

**Keywords:** children, food knowledge, food preferences, healthy program, obesity, overweight, physical activity, preschoolers, prevention, waist circumference

## Abstract

Weight and body mass index (BMI) changes appear to be poor measures for assessing the success of most pediatric obesity prevention programs (POPP). The aim of this study is to evaluate the effectiveness of the preschool-age prevention program (3P) in improving and maintaining overtime preschoolers’ knowledge/preferences about healthy nutrition and physical activity (PA), and the relationship between acquired healthy behaviors and anthropometrics including waist circumference (WC). Twenty-five preschoolers underwent a 24-month healthy lifestyle multi-component pilot intervention followed by a one-year wash-out period; 25 age-matched served as controls. Anthropometric/behavioral data were monitored. After the 2-year study and wash-out, the rates of children overweight and with obesity decreased only in the intervention group, where, also, normal-weight children with visceral obesity attained WC normal values (*p* = 0.048). While mean values of BMI Z-scores remained unchanged in both the intervention and control groups, WC (values and percentiles) showed a significant reduction only in the intervention group. Children’s adherence to the Mediterranean diet remained acceptable among the entire sample. Although daily sweet beverage consumption remained unchanged in both groups, knowledge/preferences improved significantly more in the intervention group. In conclusion, WC may be more sensitive than BMI for monitoring preschoolers in POPP and reflects healthy behavioral changes acquired during the intervention.

## 1. Introduction

The prevalence of obesity in the pediatric population has almost tripled in recent decades [1]. Among adolescents with obesity, the most rapid weight gain occurs between 2 and 6 years of age; most children who had this condition at that age maintain it in their teens [2] and this condition frequently endures into adulthood [3]. Early prevention [4] and combined dietary/physical activity (PA) behavioral interventions [5] have been proposed as a leading strategy to limit the obesity epidemic.

Weight loss and body mass index (BMI) improvement unfortunately appear to be poorly suitable measures of the efficacy of interventions performed in school/pre-school settings. It is unclear whether this finding is due to the short duration of interventions or to their late initiation after lifestyle and dietary habits have already been well established [5,6] or to the selection of inadequate outcomes (e.g., reduction of BMI rather than waist circumference (WC)) [7]. 

Strong evidence supports developmental and lifestyle factors as influential in obesity risk in infants and preschool children [8]. Preschool-age (3–6 years) studies suggest that this period of developmental plasticity, rapid weight gain, and habit formation could be a “window of opportunity” for prevention programs [2,9,10,11]. In fact, they might delay the adiposity rebound [12] associated with the increased BMI and fat mass index in adolescence [13]. Nonetheless, effective strategies in this younger age group still appear largely inconclusive [14,15,16].

Here, we evaluate the effectiveness of the preschool-age prevention program (3P) not only for improving but also for maintaining preschoolers’ knowledge/preferences regarding PA and healthy nutrition. We also evaluate the relationship between acquired healthy behaviors and specific anthropometric parameters outcomes and try to identify possible useful outcomes to measure efficacy of prevention programs in preschool-age.

## 2. Materials and Methods

The 3P is a multi-component pilot prevention program developed by a collaboration between the University and the Municipality of Salerno, a medium-income town of the Campania region in southern Italy. A research team formed by a pediatrician, a dietician, and a licensed youth exercise coach designed the study protocol, which consisted of a baseline investigation, daily classroom activities, and four follow-up evaluations.

Fifty preschoolers (mean age at entry: 3.4 ± 0.4 years) attending two public kindergartens located in comparable average social, economic, and cultural districts of Salerno were enrolled in this pilot prospective long-term study after obtaining written informed parental consent. The study was conducted in accordance with the Declaration of Helsinki, and the protocol was approved by the local institutional Ethics Committee.

Twenty-five children frequenting a class randomly selected in one of the two schools were the intervention group. Another twenty-five children frequenting a class randomly selected in the second school were assigned to the control group. Before the intervention, children and their families underwent anthropometric and/or anamnestic investigations, which were reassessed at 6, 12, and 24 months after the start of the intervention. A last follow-up evaluation was performed 12 months after the end of intervention classroom activities.

Forty-five preschoolers completed the study; five children (two in the intervention group and three controls) were excluded because they could not participate in all evaluations.

### 2.1. Baseline and Follow-Up Investigations

A medical school student, a pediatric resident, and a research nutritionist performed anthropometric measurements and anamnestic data collection. A calibrated scale (100 g precision), stadiometer (0.5 cm precision) and flexible tape (0.1 cm precision) were used to measure height, weight, and waist circumference (WC). Shoes and heavy clothing were removed before the measurement. BMI (kg/m^2^) classes were determined according to the WHO gender- and age-adjusted BMI percentiles. We considered the 85th and 95th percentiles as cut-offs for overweight (OW) and obesity, respectively [17]. The WC was measured at the end of a natural breath, midway between the iliac crest and the lower palpable rib. The visceral fat cut-off was a WC at or above the 90th percentile [18]. 

A game-interview, based on the “Food and Activities Test” [19], was used to explore knowledge and preferences about PA and nutrition. The examiner invited the children to choose among 15 pairs of images, representing healthy/unhealthy food and indoor/outdoor activities (Table 1). The interviewer asked each child to point to the food and activity that he/she personally liked best (to determine the preference parameter). Then, the examiner presented the same pairs of images and asked the child which food/activities would help them to grow healthy and strong (to determine the knowledge parameter). 

Furthermore, the interviewer investigated the habits of families and children through a parental self-reported questionnaire [20]. Children’s adherence to a Mediterranean diet (MD) (fish, vegetables, legumes, fruits, whole-grain products, extra-virgin olive oil, and yogurt) was evaluated with the MD quality index (KidMed) [21] as follows: ≥8 high, 4–7 medium, and ≤3 poor. Multiple-choice questions explored children’s consumption of sweetened beverages and fruits/vegetables. PA was recorded as mean time per week spent in non-sedentary activities (playing outside, running, cycling, etc.). Sedentary behaviors were evaluated from the parental report of “screen time” (weekly mean time preschoolers spent watching TV, playing videogames, and using a computer/smartphone) [22].

### 2.2. Classroom Activities

Before the start of classroom activities, teachers of the intervention school have attended 3 seminars/meetings with medical/nutritional staff and a licensed youth exercise coach. During the first 24 months, the intervention group performed classroom activities consisting of coloring booklets containing nutritional tips, Mediterranean diet-based recipes, and information about the benefits of PA, discussed with teachers and parents, who were invited to replicate the proposed advices and recipes at home; weekly healthy snacks distributed by local agricultural and dairy companies, consisting of sliced fresh fruit, and low-fat and sugar-free yogurt to help children overcome their usual food neophobia through repeated exposures to these healthy foods; daily PA sessions (60 min/day), supervised once a week by an expert trainer and, in remaining week days, directly by teachers. Parents were involved in periodical school meetings (3 for each year of intervention) centered on discussing childhood overweight/obesity and their complications, nutrition principles, and the Mediterranean diet and PA as tools for prevention, and participate monthly in a healthy lunch with their children at the school canteen. During the next 12 months, after the end of intervention, children continued their usual kindergarten curriculum without new additional measures focused on healthy food and PA (wash-out period).

The control group continued the usual kindergarten curriculum during the complete study period (36 months).

As the study participants were not selected for obesity, the primary outcome of the program was the improvement and maintenance over time of preschoolers’ knowledge/preferences regarding PA and healthy nutrition. The secondary outcome was the existence of a relationship between acquired healthy behaviors and specific anthropometric parameters outcomes.

### 2.3. Statistical Analysis

Data are reported as the percentage distribution or mean ± standard error. The statistical analysis was conducted with GraphPad Prism 7.03 (GraphPad Software, Inc., California, USA) and R software 3.4.1 (R Foundation for Statistical Computing, Vienna, Austria. URL http://www.R-project.org/). At the follow-up points, the two groups were compared using a paired *t*-test. Repeated-measures analysis of variance (ANOVA) was conducted to compare each variable between the two groups. Collected data were also analyzed by linear regression to examine the association between PA and healthy food knowledge/preferences (continuous independent variable) and anthropometric measures (outcome continuous variables: BMI and waist circumference). Statistical significance was indicated by *p* < 0.05.

## 3. Results

### 3.1. Anthropometric Changes

Overall, no adverse events were reported. Distribution of intervention and control children according to weight classes are shown in Table 2. After the two-year study and a one-year wash-out period, the intervention group’s overweight and obesity rates decreased, and the number of normal-weight children with visceral obesity also showed a WC reduction to normal values (50th–75th percentile). In the control group, the data collected remained unchanged. Nonetheless, as shown in Table 3, only the WC percentiles showed a significant reduction in the intervention group (*p* < 0.0001) whereas the BMI Z-score values remained unchanged in both.

### 3.2. Behavioral Changes 

Knowledge and preferences regarding healthy foods and PA, and screen time and PA levels showed a significant improvement only in the intervention group. Children’s adherence to Mediterranean diet remained acceptable among the entire sample although daily sweet beverage consumption remained unchanged in both groups as well (Table 3).

### 3.3. Anthropometric/Behavioral Relations

In contrast to the control group, preschoolers in the intervention group showed close relationships between anthropometric and behavioral parameters, with significant relations between PA knowledge and preferences (*p* = 0.0002, *r*² = 0.5033) and between healthy food knowledge and preferences (*p* = 0.0252, *r*² =0.2166). In both the intervention group and control group, the WC percentile did not correlate with knowledge or personal preferences of healthy food. The WC percentile showed a statistically significant trend towards a correlation with PA preferences (*p* = 0.051, *r*² = 0.1695) and knowledge (*p* = 0.050, *r*² = 0.2737) only in the intervention group (Figure 1).

## 4. Discussion

The 3P prevention program evaluated a sample of preschoolers monitored from kindergarten until primary school.

Implementation of the 3P program appears to be effective in some areas. Similar to previous pediatric obesity prevention studies [14,15,16,23,24,25,26], in the intervention group, an overall statistically significant reduction of BMI was not observed. Conversely, children involved in the program showed a significant reduction of visceral fat, as did those with normal weight who initially had WC values at or above the 90th percentile, whereas the control group data remained unchanged over time.

Regarding the skills that were acquired and improved during the preschool prevention program, we found an increase in their PA and food knowledge and preferences, as assessed by the Food and Activities Test [19]. This is in keeping with the idea that nutrition interventions may be more effective in helping children to make healthy food choices if developmental limitations in preschoolers’ abilities to categorize food are addressed in their class activities [27]. In fact, even though preschoolers may be unable to associate eating healthy with future health, by amassing correct information in the early years, they may be better prepared for making this connection as they gradually mature [28].

Although PA levels and screen time were significantly improved by preschool intervention and the information booklet given to preschoolers’ families, adherence to a Mediterranean diet and sweet beverage consumption instead remained unchanged. This is in keeping with the logical assumption that behavior-based programs need to be combined with the community-based prevention focused on family and home feeding practice to counteract the “obesogenic” environment [29,30,31,32].

### 4.1. Methodological Observations

These encouraging results allow us to make considerations about measurement tools and methodological drafting of prevention programs involving preschoolers. In particular, the role of the BMI, a parameter that has long been considered the reference tool to assess the success of interventions, appears controversial, likely because a lack of improvement is not always clearly related to failure to adopt a healthier lifestyle [27]. While BMI assessment may be appropriate to identify populations with a high prevalence of obesity that could benefit from healthy-lifestyle programs (e.g., low-income families) [25,33], the perspective improvement of preschoolers’ BMI may be less informative when referred to average-income families as in the present study without high prevalence of early obesity. In contrast, as suggested by this study, reduction of WC, a measurable outcome that is rarely considered in preventive interventions [14,15,23], may be more suitable for these early-aged children.

Although improved food and PA knowledge acquired during the intervention correlated with their respective preferences, only PA appeared to influence the WC percentile. These results may be incorporated into a model in which one of the objective efficacy issues of the intervention (WC reduction) is influenced at this age by PA rather than food. This finding, which has been rarely explored and reported in previous preschool-age interventions, warrants further studies.

### 4.2. Strengths and Limitations of the Study

Limitations of our pilot study include the small sample size and the use of parent self-reported data. In fact, newer and more objective techniques to measure PA [34] and nutrition knowledge [35,36,37] should ideally be adopted in the future and might be worth as well to confirm our results [38]. In this regard, recently, Wiseman et al. reported data using this test enhanced by a user-friendly electronic device [39]. Nevertheless, the game interview “Food and activities test” [19] appears to be child appealing, easy to perform, and adaptable in different cultural and social contexts. Furthermore, the direct family involvement through home-visiting programs also recently emerged as an effective tool to decrease the rate of childhood obesity [29,30,40,41], sustaining the fact that a combined approach starting early in life might be more useful rather than school-based intervention alone [7].

Strengths of our study possibly reside in involvement of an ethnically homogeneous population of average income, the adequately long duration of classroom activities (24 months), and the evaluation of a variety of parameters during a long-term follow-up aiming to verify maintenance of acquired progress after a wash-out period (12 months). All these aspects have rarely been considered in other studies [14,15,16]. Our proposed time protocol (long intervention plus wash-out period) seems to be promising to define the over-time efficacy of similar preschool projects. A longer-term follow-up with a longer wash-out period could be useful to evaluate the efficacy of preschool programs in retaining the early acquired knowledge/preferences later in life (i.e., school-age, adolescence, adulthood), and should be considered in future similar studies. Lastly, as suggested by a recent systematic review, the assessment of academic and cognitive outcomes should be considered in future obesity prevention studies, considering the presence of possible benefits of similar interventions also on these topics [42].

## 5. Conclusions

An early and adequately long obesity prevention program can influence pediatric behavior and possibly encourage the acquired healthy lifestyle information to be retained later in life.

Our results suggest that it is necessary to combine behavioral and environmental parameters with knowledge acquired in a pre-school context to reach an improvement of anthropometrics. Even if further studies need to explore this field, WC appears to be an easily measurable parameter to be monitored to assess compliance to healthy lifestyle and seems to be more influenced by PA rather than food in preschool years. 

## Figures and Tables

**Figure 1 nutrients-11-02139-f001:**
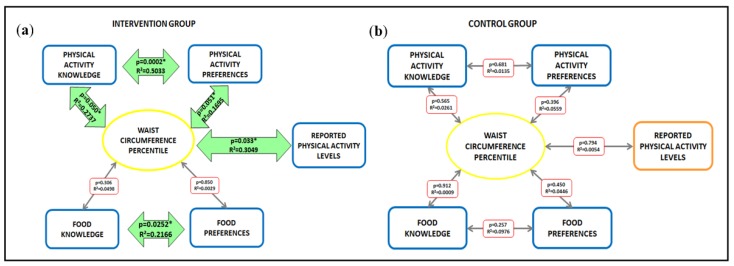
Linear regression examining the association between physical activity and healthy food knowledge/preferences and anthropometric measures in intervention (**a**) and control groups (**b**). Green arrows and ***** indicate statistically significant relationship (*p* < 0.05).

**Table 1 nutrients-11-02139-t001:** Pairs of images used in the “Foods and Activities Test” adapted from Calfas et al. [19].

Healthy Choices	Unhealthy Choices
Foods	
1. Rice	French fries
2. Fruits	Chocolate snacks
3. Boiled fish	Sausages
4. Ice lolly	Ice cream
5. Orange juice	Cola
6. Peanuts	Popcorn with butter and salt
7. Whole grain biscuits	Potato chips
8. White yogurt	Spread chocolate cream
Activities	
1. Dancing	Coloring
2. Swimming	Watching TV
3. Sliding	Swinging
4. Bicycling	Sitting and rolling toy train
5. Kicking ball	Playing videogames
6. Climbing on jungle gym	Playing with Legos
7. Running	Walking

**Table 2 nutrients-11-02139-t002:** Body mass index (BMI) and waist circumference characteristics in intervention (*n* = 23) and control (*n* = 22) groups at baseline (T0) and last follow-up = end of wash-out (T4).

	Intervention Group (n = 23)			Control Group (n = 22)		
	NW n (%)	OW n (%)	OB n (%)	NW n (%)	OW n (%)	OB n (%)
Baseline (T0)	16 (69.5%) (5 (21.7%))•	3 (13%)	4 (17, 5%)	18 (82%) (7 (31.8%))•	2 (9%)	2 (9%)
Follow-up (T4)	19 (82.6%) (0 (0%))•	2 (8, 7%)	2 (8, 7%)	18 (82%) (7 (31.8%))•	2 (9%)	2 (9%)
EFT *p*-value	ns (0.0486)•	ns	ns	ns (ns)•	ns	ns

BMI: Body mass index; ns: Not statistically significant; NW: Normal weight; OW: Overweight; OB: Obesity; EFT: Exact Fisher test between T0-T4; •: NW subjects with visceral obesity.

**Table 3 nutrients-11-02139-t003:** Descriptive statistics, repeated-measures ANOVA results by time, and paired *t*-test of assessments of main parameters of our sample (*n* = 45).

	Intervention Group						Control Group					
	T0	T1	T2	T3	T4	*p*	T0	T1	T2	T3	T4	*p*
**Age (years)**	3.4 ± 0.4	4.0 ± 0.4	4.6 ± 0.4	5.4 ± 0.4	6.5 ± 0.6		3.4 ± 0.4	4.0 ± 0.4	4.6 ± 0.4	5.4 ± 0.4	6.5 ± 0.6	
**BMI (Z-score)**	0.05 ± 1.19	0.08 ± 1.26	0.36 ± 1.35	0.53 ± 0.95	0.42 ± 0.98	0.5382(a) 0.0830(b)	0.22 ± 1.12	0.23 ± 0.91	0.34 ± 0.98	0.12 ± 1.08	0.23 ± 1.17	0.9883(a) 0.9561(b)
**WC (percentile)**	78.9 ± 16.5	68.4 ± 22.3	62.4 ± 20.9	59.3 ± 20.4	58.6 ± 21.0	0.0049(a) <0.0001(b)	70 ± 23.5	56.0 ± 21.4	46.5 ± 23.8	45.4 ± 23.9	51.7 ± 28.1	0.9950(a) 0.5041(b)
**Food knowledge**	3.8 ± 1.6	4.8 ± 1.6	5.2 ± 1.9	6.1 ± 2.0	6.5 ± 1.1	0.0001(a) <0.0001(b)	4.1 ± 2.3	5.6 ± 2.0	6.6 ± 1.6	6.6 ± 1.6	6.4 ± 1.2	0.6349(a) 0.4806(b)
**Food preferences**	3.2 ± 1.6	4.6 ± 1.6	3.7 ± 1.7	4.8 ± 2.0	5.4 ± 1.6	<0.0001(a) 0.0007(b)	2.9 ± 2.0	4.8 ± 2.3	4.8 ± 2.2	4.6 ± 2.4	4.3 ± 1.8	0.5205(a) 0.2282(b)
**PA knowledge**	3.7 ± 1.6	3.8 ± 1.5	4.4 ± 1.5	4.9 ± 1.6	5.4 ± 1.2	0.002(a) <0.0001(b)	3.4 ± 1.9	4.1 ± 2.0	4.4 ± 1.0	4.6 ± 1.3	4.3 ± 0.9	0.8380(a) 0.5191(b)
**PA preferences**	3.0 ± 1.5	4.3 ± 1.6	4.2 ± 1.3	4.7 ± 1.4	5.3 ± 1.2	<0.0001(a) <0.0001(b)	3.1 ± 1.7	3.8 ± 1.7	4.1 ± 1.2	4.0 ± 1.3	4.6 ± 1.2	0.4398(a) 0.3012(b)
**KidMed score**	6.2 ± 2.1	6.3 ± 2.3	7.0 ± 1.9	6.7 ± 2.1	6.9 ± 2.1	0.4580(a) 0.1503(b)	4.9 ± 1.4	5.4 ± 1.3	6.0 ± 1.8	5.4 ± 2.8	6.0 ± 2.0	0.9819(a) 0.9999(b)
**Sweet beverage (mL/week)**	97.0 ± 105.0	103.0 ± 108.0	47.0 ± 77.0	70.0 ± 101.0	49.0 ± 50.0	0.4394(a) 0.1358(b)	45.0 ± 60.0	28.0 ± 36.0	30.0 ± 42.0	40.0 ± 38.0	52.5 ± 60.0	0.9599(a) 0.6793(b)
**Screen time (min/week)**	163.5 ± 46.6	147.8 ± 46.2	133.6 ± 37.1	139.6 ± 52.9	136.3 ± 50.8	0.2051(a) 0.0098(b)	151.0 ± 37.0	130.0 ± 32.9	113.3 ± 29.0	147.1 ± 31.6	156.8 ± 48.52	0.9713(a) 0.2142(b)
**PA (min/week)**	84.6 ± 36.7	89.1 ± 37.7	94.7 ± 38.1	98.4 ± 41.8	105.2 ± 40.6	0.4621(a) 0.0106(b)	84.1 ± 27.8	83.4 ± 34.9	89.8 ± 29.0	86.5 ± 34.3	91.0 ± 40.4	0.9651(a) 0.2206(b)

T0: Baseline; T1 and T2: 1st and 2nd follow up; T3: End of intervention; T4: End of wash-out; ANOVA: Analysis of variance; BMI: Body mass index; WC: Waist circumference; min: Minutes; PA: Physical activity. Values represent Means ± SD; *p*-value ANOVA (a) and *p*-value *t*-test (b) T0 vs. T4.

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
