# Peer review of "Waist Circumference and Healthy Lifestyle Preferences/Knowledge Monitoring in a Preschool Obesity Prevention Program"

_nutrients, 2019, doi:10.3390/nu11092139_

Round 1

Reviewer 1 Report

This interesting and well-written paper describes a pre-school obesity prevention program that was implemented over a period of 2 years. The authors are to be commended on the scope and design of the study as well as the developmental appropriateness of the intervention. The following comments are intended to help the authors improve upon an already strong manuscript.

Authors are encouraged to use person-first language throughout the manuscript (e.g., children with obesity vs. obese children) The role of parental caregivers in the intervention as well as assessment is not entirely clear. Please provide more specifics on how exactly parents are involved in this study. What about the roles of other adults (e.g., teachers). Retention over 3 years is impressive. Were there incentives provided? Please provide this information in methods. Please provide more clear labels for Table 1 in order to assist reader in interpretation. Labels in column headings such as n(%) will be helpful. Would there be utility in longer-term follow-up? This seems to be a missing element in the discussion / future directions.

Author Response

RESPONSE TO REVIEWER 1

This interesting and well-written paper describes a pre-school obesity prevention program that was implemented over a period of 2 years. The authors are to be commended on the scope and design of the study as well as the developmental appropriateness of the intervention. The following comments are intended to help the authors improve upon an already strong manuscript

Authors are encouraged to use person-first language throughout the manuscript (e.g., children with obesity vs. obese children)

According to the reviewer's suggestion, in the revised manuscript we use person-first language (lines 34-35)

The role of parental caregivers in the intervention as well as assessment is not entirely clear. Please provide more specifics on how exactly parents are involved in this study. What about the roles of other adults (e.g., teachers).

We specified in methods section the role and the involvement of parents and teachers.(lines 109-121)

Retention over 3 years is impressive. Were there incentives provided? Please provide this information in methods.

Activities were performed only during the first 24 months. During the next 12 months, children continued their regular kindergarten curriculum without incentives (wash-out period). We tried to specify better this concept in methods section. (lines 121-124)

Please provide more clear labels for Table 1 in order to assist reader in interpretation. Labels in column headings such as n (%) will be helpful.

As suggested we added n (%) in column headings.

Note that table numeration has changed, as we included the previou supplementary table S1 (now table 1) in the main text.

Would there be utility in longer-term follow-up? This seems to be a missing element in the discussion / future directions. 

Thanks for the suggestion. We added a sentence about the utility of longer follow-up in the section “Strengths and limitations of the study”. (lines 388-391)

Reviewer 2 Report

This is an interesting study, there has clearly been good attention paid to being diligent with the data collection.  I think that the paper suffers a little from considering both the effects of the intervention and the measurement of the changes. Each section needs to be thoroughly reviewed so that each of these is handled clearly and effectively as it is neither is quite dealt with thoroughly enough to convey the findings effectively.  The paper seems brief for the volume of data and the 2 aims and I would strongly consider dealing with these 2 things in 2 papers; one exploring the intervention in depth and a second considering methodological issues. If you want to keep it together there will be need to a systematic approach to enhancing teh clarity and dealing with each aim in a purposeful manner. 

Author Response

RESPONSE TO REVIEWER 2

This is an interesting study, there has clearly been good attention paid to being diligent with the data collection. 

I think that the paper suffers a little from considering both the effects of the intervention and the measurement of the changes. Each section needs to be thoroughly reviewed so that each of these is handled clearly and effectively as it is neither is quite dealt with thoroughly enough to convey the findings effectively. 

We appreciate Reviewer’s suggestion and modified accordingly the presentation of our results, splitting them clearly in 3 separated sections (anthropometrics changes, behavioral changes, and anthropometric/behavioral relations). (see Results sections # 3.1, 3.2, 3.3 )

We also reviewed the Discussion section by separating effects of intervention from considerations regarding measurements of efficacy.

The paper seems brief for the volume of data and the 2 aims and I would strongly consider dealing with these 2 things in 2 papers; one exploring the intervention in depth and a second considering methodological issues. If you want to keep it together there will be need to a systematic approach to enhancing the clarity and dealing with each aim in a purposeful manner. 

We tried to enhance clarity, maintaining the one paper structure (see above)